# Effects of Single and Combined Drought and Salinity Stress on the Root Morphological Characteristics and Root Hydraulic Conductivity of Different Winter Wheat Varieties

**DOI:** 10.3390/plants12142694

**Published:** 2023-07-19

**Authors:** Yuanyuan Fu, Penghui Li, Abdoul Kader Mounkaila Hamani, Sumei Wan, Yang Gao, Xingpeng Wang

**Affiliations:** 1Institute of Farmland Irrigation, Chinese Academy of Agricultural Sciences, Xinxiang 453002, China; fyy2016060105@163.com (Y.F.); m_abdoulkader@yahoo.com (A.K.M.H.); 2College of Agronomy, Tarim University, Alar 843300, China; wansumei510@163.com; 3College of Water Conservancy and Architecture Engineering, Tarim University, Alar 843300, China; 13999068354@163.com

**Keywords:** root hydraulic conductivity, combined PEG and NaCl stress, PEG stress, salt stress, protective enzyme activities

## Abstract

Water shortages and crop responses to drought and salt stress are related to the efficient use of water resources and are closely related to food security. In addition, PEG or NaCl stress alone affect the root hydraulic conductivity (Lpr). However, the effects of combined PEG and NaCl stress on Lpr and the differences among wheat varieties are unknown. We investigated the effects of combined PEG and NaCl stress on the root parameters, nitrogen (N) and carbon content, antioxidant enzymes, osmotic adjustment, changes in sodium and potassium, and root hydraulic conductivity of Yannong 1212, Heng 4399, and Xinmai 19. PEG and NaCl stress appreciably decreased the root length (RL), root surface area (RS), root volume (RV), K^+^ and N content in shoots and roots, and Lpr of the three wheat varieties, while the antioxidant enzyme activity, malondialdehyde (MDA), osmotic adjustment, nonstructural carbon and Na^+^ content in shoots and roots, etc., remarkably remained increased. Furthermore, the root hydraulic conductivity had the greatest positive association with traits such as RL, RS, and N and K^+^ content in the shoots of the three wheat varieties. Moreover, the RL/RS directly and actively determined the Lpr, and it had an extremely positive effect on the N content in the shoots of wheat seedlings. Collectively, most of the root characteristics in the wheat seedlings decreased under stress conditions, resulting in a reduction in Lpr. As a result, the ability to transport nutrients—especially N—from the roots to the shoots was affected. Therefore, our study provides a novel insight into the physiological mechanisms of Lpr.

## 1. Introduction

The adverse effects of single or multiple environmental stresses, such as drought, salinity, high and low temperatures, nutrient deficiencies, etc., on plant growth and yield have become increasingly severe in recent years due to increased global climate change and the occurrence of extreme weather events [1,2,3,4,5,6,7]. Moreover, the increasingly severe salinization in arid and semi-arid areas reduces crop yields and threatens global food security [8].

The effects of drought stress and salinity stress on the physio-biochemical properties of plants are very similar. Both are harmful to plants through osmosis, toxicity, and the accumulation of reactive oxygen species (ROS), resulting in developmental changes, metabolic restriction, ion isolation or rejection, and growth inhibition in plants [1,9]. During drought stress and salt stress, the active accumulation of solutes such as soluble carbohydrates, proteins, and free amino acids is considered to be an effective mechanism for resisting stress. The production of ROS results in an increase in the lipid peroxidation level; the malondialdehyde (MDA) is a lipid peroxidation product that is used as a biochemical marker of oxidative stress. Moreover, the scavenging of ROS and protection from oxidative damage are also components of a plant’s self-defense system. ROS scavenging systems consist of enzymatic and non-enzymatic scavenging systems. The enzymes used to defend against oxidative damage include superoxide dismutase (SOD), peroxidase (POD), catalase (CAT), and so on. The non-enzymatic scavenging systems contain glutathione (GSH), ascorbic acid (AsA), proline, etc. [10]. However, the difference between the two types of stress is that drought stress induces the accumulation of osmotic compounds, while salt stress causes the synthesis of compatible solutes and regulates ion migration in plants [11]. Additionally, salt stress inhibits plant growth through the specific effects of ions—mainly Na^+^ and Cl^−^ [1,12,13]. The accumulation of ionic content (e.g., K^+^, Na^+^, and Cl^−^) disturbs ionic homeostasis, resulting in cell membrane dysfunction, the weakening of metabolic activity, and a further reduction in growth [12]. In all, drought- and salinity-induced osmotic stress limits plants’ uptake of water and nutrients, resulting in reduced initial growth. The reduced area for photosynthesis in plants minimizes the accumulation of photosynthetic products, ultimately leading to lower yields. Therefore, restricted water and nutrient uptake in crops is one of the main reasons for the decreased yield under the environmental stresses of drought and salinity [14,15].

In the soil–plant–atmosphere continuum, roots supply water and soil nutrients to the parts of the plant above the ground [16]. When plants are subjected to drought and salt stresses, the root system is the first organ to sense and respond to them. The root hydraulic conductivity represents the water uptake and transport capacity, and it might control the hydraulic conductance and vigor of the whole plant [17,18,19]. Root hydraulic conductivity is related to the root system and root morphology. It depends on the root surface area, anatomy, and water permeability of the roots of the plant [20,21,22]. According to research findings, the root hydraulic conductivity is not only regulated by aquaporin, but is also related to root colonization, which plays a critical role in water uptake [23,24]. In recent years, an increasing body of work has linked plant roots’ water uptake capacity and water transport across membranes to hydraulic conductivity [25,26]. There have been several reports on the influence of environmental factors, such as soil mechanical resistance, soil bulk density, drought, salinity, and soil ventilation—all of which will affect root hydraulic conductivity—on root water absorption [20,27,28]. Previous results showed that drought stress reduced the root hydraulic conductivity of cedar [29], eucalyptus [30], citrus [31], tomato [32], rice [33,34], and corn [35]. Salt stress can also reduce the root hydraulic conductivity of saltbush [36], rice [26], *Beta vulgaris* [37], tomato [1], and soybean. In addition, several experiments revealed that the allocation of non-structural carbohydrates is essential for plants’ hydraulic recovery under drought conditions [38,39]. Recently, Wang et al. [7] found that plants did not respond uniformly to drought, salt, and dual-stress treatments at the physiological and transcriptional levels. However, little information is available on the influence of co-occurring drought and salt stress on the root hydraulic conductivity of plants.

Therefore, it is necessary to sufficiently understand the plants’ physiological responses to combined drought and salt stress and, particularly, the changes in the root system and root hydraulic conductivity in order to improve the tolerance of crops to drought and salt stress and, thus, guarantee world food security. In this work, three wheat varieties with significant differences in drought and salt tolerance were cultivated with a hydroponic method, and polyethylene glycol (PEG-6000) and NaCl were added to the nutrient solution to simulate environments of drought stress, salt stress, and combined drought–salt stress. In addition, these treatments started from the seed germination stage. Considering the sensitivity of wheat seeds to salinity, the measurement of the Lpr, and the cumulative effect [40,41], we used the 0.1% NaCl concentration in the present study. This study was designed to analyze the root parameters, root hydraulic conductivity, osmotic regulation abilities, antioxidant enzyme activity, and contents of nitrogen, non-structural carbohydrates, potassium, and sodium of wheat under these three stress conditions. Furthermore, the roots’ morphological parameters and physio-biochemical responses to root hydraulic conductivity were investigated in the study. Our objective was to investigate the relationship between root hydraulic conductivity and the physiological characteristics of Yannong 1212, Heng 4399, and Xinmai 19 under PEG stress, NaCl stress, and coupled PEG and NaCl stress.

## 2. Results

### 2.1. Effects of PEG and NaCl Stress on the Root Morphological Parameters of Different Wheat Varieties

The effects of PEG and NaCl stress on the root parameters of the three wheat varieties are shown in Figure 1. The PEG (D), NaCl (S), and coupled PEG and NaCl (DS) treatments significantly affected the root morphology. The effects of stress on the root parameters of the three wheat varieties were not consistent. Compared with that of CK, the root lengths of the three wheat seedlings (Yannong 1212, Heng 4399, and Xinmai 19) were significantly reduced under D and DS stress. Under D stress, in comparison with the other treatments, the root length decreased by 55%, 33%, and 56%, respectively, while it was reduced by 96%, 92%, and 96%, respectively, under DS stress in Yannong 1212, Heng 4399, and Xinmai 19 (Figure 1a). Under S stress, the root lengths of Yannong 1212 and Xinmai 19 were reduced by 47% and 30%, respectively, in comparison with that of CK, but in Heng 4399, it was significantly increased by 64% (Figure 1a). Interestingly, in comparison with that of CK, the average root diameters of Yannong 1212 and Heng 4399 under DS stress were significantly greater than those under the other stresses, exhibiting increases of 122% and 150%, respectively (Figure 1b). However, in Xinmai 19, this value was extremely reduced under the D, S, and DS stresses in comparison with that of CK (Figure 1b). The root surface area and root volume of Yannong 1212 and Xinmai 19 were greatly reduced under the three stresses, whereas in Heng 4399, they significantly increased by 48% and 33%, respectively, under salt stress (Figure 1c,d). To summarize, compared with the control, D and DS significantly reduced the root length, root surface area, and root volume of the three wheat varieties, and DS decreased them even more (Figure 1a,c,d). However, the differences in root length, root surface area, and root volume of the three wheat varieties under the DS stress conditions were insignificant (Figure 1a,c,d). Meanwhile, the ANOVA showed that not only the stress types and the variety, but also their interaction, had a very significant impact on the root parameters of the three wheat seedlings (Figure 1).

### 2.2. Effects of PEG and NaCl Stress on the Nitrogen, Potassium, and Sodium Contents of Wheat Seedlings

As shown in Figure 2, the total nitrogen (N) in the shoots and roots of Yannong 1212, Heng 4399, and Xinmai 19 showed an almost uniform trend under the single stresses and the coupled PEG and NaCl stress. In comparison with that in CK, the total N content in the shoots of the three wheat varieties was significantly reduced under D, S, and DS, and it decreased the most under DS, reaching 43%, 55%, and 40%, respectively (Figure 2a). Nevertheless, in Yannong 1212, Heng 4399, and Xinmai 19, under S, these values were only reduced by 10%, 5%, and 6%, respectively, in comparison with that of CK (Figure 2a). By comparison, the total N content in the roots of the three wheat varieties appreciably decreased under D and DS, but there was no significant difference for that under S (Figure 2b). Similarly, the total N content in the roots of Yannong 1212, Heng 4399, and Xinmai 19 was remarkably reduced by 34%, 38%, and 33%, respectively, and it decreased the most among the three stresses in comparison with CK (Figure 2b).

The ANOVA showed that the types of stress, the variety, and their interaction had extremely significant effects on the total N content in the three wheat seedlings (*p* < 0.01) (Figure 2).

In addition, we analyzed the variations in potassium (K^+^) and sodium (Na^+^) contents in the stems and roots of the three varieties under D, S, and DS. As shown in Table 1, in comparison with that of CK, the content of K^+^ in the shoots and roots of the three wheat seedlings significantly decreased under D, S, and DS. On the contrary, the Na^+^ content and Na^+^/K^+^ in the shoot and root tissues of the three wheat varieties, remarkably, mostly increased under the three stresses. When we further analyzed the content of K^+^ in the shoots and roots of Yannong 1212, Heng 4399, and Xinmai 19 under the three stresses, we found that this value appreciably decreased the most under DS, with levels of up to 76%, 86%, and 67% in the shoots and 77%, 62%, and 70% in the roots, respectively, in comparison with that in CK. It was obvious that the content of K^+^ in the shoot and root tissues of Yannong 1212 decreased the most. Moreover, the content of Na^+^ in the shoots of Yannong 1212, Heng 4399, and Xinmai 19 under DS increased the most, reaching by 8-fold, 16-fold, and 7.5-fold changes, respectively, in comparison with that in CK. However, by contrast, the content in the roots of Yannong 1212, Heng 4399, and Xinmai 19 under salt stress alone increased by 9.2-fold, 7-fold, and 3.8-fold, respectively, thus representing the greatest increase. D had no remarkable effects on the Na^+^/K^+^ ratios in the shoots and roots of the three wheat varieties.

However, S and DS markedly increased the Na^+^/K^+^ ratios in all of the tissues of the three wheat varieties that were measured, and DS caused the greatest increase. Furthermore, the stress types (S), different varieties (V), and S × V interaction had extremely significant effects on the K^+^ content, Na^+^ content, and Na^+^/K^+^ ratios of the three wheat seedlings (*p* < 0.01).

### 2.3. Effects of PEG and NaCl Stress on Non-Structural Compounds

The effects of PEG, NaCl, and coupled PEG and NaCl stress on non-structural carbohydrates in the wheat seedlings are shown in Figure 3. Yannong 1212 had the greatest total starch content in the shoots and roots among the three wheat varieties when under DS (Figure 3a). This value increased under D and DS, but was reduced under S in Yannong 1212 and Xinmai 19 (Figure 3a). However, this value increased under D, S, and DS in Heng 4399 (Figure 3a). Moreover, compared with that in CK, the total content in the three wheat varieties was the greatest when under DS (Figure 3a). For the starch content in the shoots, the trend of variation for the three varieties was consistent with that for the total starch content (Figure 3a). The trends of variation in the content in the roots in the three varieties were different (Figure 3a). In Yannong 1212, compared with CK, this value decreased by 8% and 19%, respectively, under D and S, but it significantly increased by 73% under DS (Figure 3a). In Heng 4399, it increased remarkably under D, S, and DS by 50%, 15%, and 177%, respectively, in comparison with CK (Figure 3a). In Xinmai 19, by comparison, there were no significant changes under D and S, but this value appreciably increased by 58% under DS (Figure 3a).

Moreover, we analyzed the fructose and sucrose contents in the shoots and roots of the three wheat varieties. The total fructose content and the total sucrose content were the greatest in Heng 4399 when under DS and were the lowest in Xinmai 19 when under S (Figure 3b,c). For the fructose content in the shoots, in both Yannong 1212 and Xinmai 19, it was insignificant, but in Heng 4399, it remarkably increased by 52% under D in comparison with CK (Figure 3b). Under S, compared with CK, this value was significantly reduced in Yannong 1212, but it increased markedly in Heng 4399, and there were no significant differences for Xinmai 19 (Figure 3b). Additionally, under DS, in Yannong 1212, Heng 4399, and Xinmai 19, this value appreciably increased by 204%, 215%, and 90%, respectively, in comparison with that for CK (Figure 3b). The fructose in the roots was reduced under D and S, but it significantly increased under DS in Yannong 1212 and Xinmai 19. It increased by 67% and 212%, respectively, under D and DS, but it decreased by 26% under S in Heng 4399 in comparison with that in CK (Figure 3b). For the sucrose content in the shoots, compared with CK, there were no significant differences under D and S, but this value appreciably increased by 203% and 152%, respectively, under DS in Yannong 1212 and Xinmai 19 (Figure 3c). However, in Heng 4399, it significantly increased under D, S, and DS compared with that in CK by, respectively, 83%, 64%, and 318% (Figure 3c). As for the content in the roots, compared with that in CK, there were no significant differences under D, but this value appreciably increased by 302% and 135%, respectively, under DS in Yannong 1212 and Xinmai 19 (Figure 3c). However, in Heng 4399, it increased extremely under both D and DS by, respectively, 222% and 682% in comparison with that in CK (Figure 3c). Under S, the content in the roots of Yannong 1212 was remarkably reduced by 48%, while that in Heng 4399 and Xinmai 19 was not significantly different from that in CK (Figure 3c).

Meanwhile, the ANOVA showed that the stress types, varieties, and their interaction had a significant (*p* < 0.01) impact on the content of unstructured carbohydrates in the wheat seedlings.

### 2.4. Effects of PEG and NaCl Stress on Malondialdehyde, Protective Enzymes, and Osmoregulatory Substances in Wheat Seedlings

MDA is a product of membrane lipid peroxidation, and the magnitude of its value can represent the degree of damage to a plant. As shown in Table 2, compared with that in CK, the MDA content in the roots of Heng 4399 increased the most under S, reaching up to 88%. Under D, in the shoots of Heng 4399 and Xinmai 19, it increased the most among the three types of stress by up to 14% and 26%, respectively, in comparison with CK. Meanwhile, in comparison with CK, the MDA content in the shoots of Yannong 1212 and the content in the roots of Heng 4399 increased the most under the combined stress of PEG and NaCl (20% and 174%, respectively). According to a further analysis, in the shoots of Yannong 1212, there were no significant differences under D and S, while the content markedly increased by 21% under DS in comparison with that in CK. In the roots of Yannong 1212, it significantly decreased by 23% under D, while it increased by 24% under S in comparison with that in CK. However, there were no significant differences under DS with respect to CK. By contrast, in the shoots of Heng 4399, the content appreciably increased by 14% and 8%, respectively, under D and DS, while it was significantly reduced by 10% under S. However, in the roots of Heng 4399, it remarkably increased by 52%, 88%, and 174%, respectively, under D, S, and DS in comparison with that in CK. The trend of the change in the shoots and roots of Xinmai 19 was different from that for Yannong 1212 and Heng 4399. There was a marked increase in the shoots under D and S by 26% and 17%, respectively, in comparison with CK. However, there were no significant differences between CK and DS in the shoots of Xinmai 19. In the roots of Xinmai 19, there were no differences under D and DS, but there was a significant increase under S. Additionally, significant differences were detected in the stress types (S), different varieties (V), and S × V interaction for the content of MDA in shoots and roots (*p* < 0.01).

As shown in Figure 4, the antioxidant enzyme (SOD, POD, and CAT) activity of wheat seedlings did not uniformly change under all of the treatment conditions. On the whole, the SOD and POD activity in the whole seedlings of Yannong 1212, Heng 4399, and Xinmai 19 was the highest under DS in comparison with that under all other treatment conditions. The CAT activity in Yannong 1212 and Heng 4399 was also the highest under DS, but that in Xinmai 19 under D was the highest among all treatment conditions. For the SOD activity in the shoots, compared with CK, D and DS, increased it by 31% and 30%, respectively, but S significantly reduced it by 12% in Yannong 1212 (Figure 4a). However, S and DS both caused an extreme increase in Heng 4399 (49% and 76%) and Xinmai 19 (17% and 117%) in comparison with CK (Figure 4a). However, there were no significant differences under D for the value in Heng 4399 in comparison with that in CK (Figure 4a). The SOD activity in the root tissues of Yannong 1212, Heng 4399, and Xinmai 19 increased by 51%, 210%, and 215%, respectively, under D in comparison with that in the control (Figure 4a). Under S, compared with that in CK, the activity in the roots of Yannong 1212 had no significant differences, while in Heng 4399 and Xinmai 19, it markedly increased by 295% and 133%, respectively (Figure 4a). DS appreciably increased the activity in the roots of Yannong 1212 and Heng 4399 by 392% and 771%, respectively, while this stress had no statistically significant effects on that in Xinmai 19 in comparison with that in CK (Figure 4a). As for the POD activity, under D, compared with CK, in the shoots of Yannong 1212, there was an extreme reduction by 28%, but in Heng 4399, it markedly increased by 40%, and that in Xinmai 19 was not significantly different (Figure 4b). It was found that, under S, in comparison with that in CK, there were no significant differences in the activity in the shoots of Yannong 1212 and Xinmai 19, whereas that in Heng 4399 was markedly increased by 54% (Figure 4b). In addition, DS appreciably increased the activity in the shoots of Yannong 1212, Heng 4399, and Xinmai 19 by 226%, 400%, and 451%, respectively (Figure 4b). For the POD activity in the roots, D caused a remarkable decrease of 12% in Yannong 1212 but had no statistically significant effects on that in Heng 4399 (Figure 4b). Nevertheless, S and DS both caused extreme increases in Yannong 1212 (13% and 36%) and Heng 4399 (143% and 162%) (Figure 4b). In addition, in Xinmai 19, the activity markedly increased under all treatment conditions, and it increased the most under S (71% in comparison with CK) among the three types of stress (Figure 4b). Furthermore, we researched the CAT activity in the shoots and roots of all of the wheat varieties. Firstly, in the shoots, compared with CK, the activity in Yannong 1212 and Xinmai 19 exhibited no significant differences under D, but it was appreciably reduced by 45% and 27%, respectively, under DS (Figure 4c). S caused a reduction in the activity in the shoots of Yannong 1212, but caused no significant differences in that of Xinmai 19 in comparison with that in CK (Figure 4c). For that of Heng 4399, compared with that of CK, D and DS markedly increased the value by 32% and 16%, respectively, but S had no significance (Figure 4c). Then, in the roots, compared with that in CK, that in Yannong 1212, Heng 4399, and Xinmai 19 significantly increased under both D and DS, and that in Yannong 1212 and Xinmai 19 decreased, but in Heng 4399, it appreciably increased under S (Figure 4c). Coincidentally, the results of the analysis of variance showed that the stress types and varieties, as well as their interaction, had an extremely significant effect on the activity of anti-oxidase (*p* < 0.01).

Table 3 demonstrates the effects of PEG and NaCl stress on osmoregulatory substances in the different varieties of wheat seedlings. Compared with that in CK, the content of soluble protein in the shoots of Xinmai 19 and in the roots of Yannong 1212 showed no differences under D, S, and DS. In the shoots, the content was not significantly different in Yannong 1212, but it was remarkably reduced by 21% in Heng 4399 under D. It appreciably increased by 54% in the shoots of Yannong 1212, while there were no differences in that of Heng 4399 under DS in comparison with CK. In addition, there were no significant differences in the shoots of Yannong 1212 and Heng 4399 under S. For the content in the roots, compared with that in CK, there were no differences in Heng 4399 and Xinmai 19 under D, but it decreased in both of these wheat varieties under S (56% and 35%, respectively). However, under DS, it was significantly reduced by 32% in the roots of Heng 4399, but it was increased by 35% in Xinmai 19. Then, we analyzed the soluble sugar content in the shoots and roots of the three wheat varieties. In Yannong 1212, Heng 4399, and Xinmai 19, the content in the shoots and roots increased the most under DS in comparison with CK. Under D, the content in the shoots of Yannong 1212, Heng 4399, and Xinmai 19 appreciably increased by 24%, 111%, and 36%, respectively, in comparison with that in CK. Under S, in comparison with that in CK, the soluble sugar content in the shoots of Yannong 1212 was reduced by 18%, and that of Heng 4399 increased by 61%. However, that of Xinmai 19 was not significantly different from that of CK. For the soluble sugar content in the roots, compared with that in CK, in Yannong 1212, it increased by 11% and 227%, respectively, under D and DS, but it was significantly reduced by 12% under S. Nevertheless, the content in the roots of Heng 4399 remarkably increased by 182%, 29%, and 243%, respectively, in comparison with that in CK. The trend in Xinmai 19 was almost the same as that in Heng 4399, but there was still a difference between them. That is, there were no significant differences in the increases under D and S in Xinmai 19. We also measured the proline content in the shoots and roots of the three wheat varieties. Compared with that in CK, the content in the shoots of Yannong 1212, Heng 4399, and Xinmai 19 markedly increased under the three types of stress—without exception—and increased the most under DS (14,599%, 5274%, and 21,634%, respectively). For the content in the roots, compared with that in CK, it significantly increased in Yannong 1212 and Heng 4399 under D and DS, but it was appreciably reduced under S. That in Xinmai 19 was increased under the three types of stress in comparison with that in CK. Overall, the same was found in Yannong 1212, Heng 4399, and Xinmai 19, which underwent the greatest increase under DS in comparison with CK (1413%, 1110%, and 839%, respectively). Moreover, the results of the analysis of variance showed that the stress types and varieties, as well as their interaction, had an extremely significant effect on the content of osmoregulatory substances (*p* < 0.01).

### 2.5. Effects of PEG and NaCl Stress on the Hydraulic Conductivity of the Root System

It can be seen in Figure 5 that PEG stress, NaCl stress, and combined PEG and NaCl stress all significantly reduced the root hydraulic conductivity and theoretical hydraulic conductivity of the wheat seedlings. Under the CK conditions, Xinmai 19 had the highest theoretical root hydraulic conductivity, and Heng 4399 had the highest root hydraulic conductivity per plant (Figure 5). Compared with the control, the theoretical root hydraulic conductivity in Yannong 1212, Heng 4399, and Xinmai 19 significantly decreased under D, S, and DS, and in Xinmai 19, it was reduced by 99% under the combined stress of PEG and NaCl (Figure 5a). There were no differences in the theoretical root hydraulic conductivity in Yannong 1212 among the three types of stress (Figure 5a). For the theoretical root hydraulic conductivity in Heng 4399, there were no significant differences between CK and D or between S and DS (Figure 5a). Interestingly, in Xinmai 19, it was reduced by 77% under D, while it decreased by 67% under S compared with that in CK (Figure 5a). Meanwhile, we measured and analyzed the root hydraulic conductivity per plant for the three wheat seedlings under the three types of stress. As shown in Figure 5b, in Yannong 1212, Heng 4399, and Xinmai 19, under the three types of stress, there was a remarkable reduction, and the hydraulic conductivity of the root system in each plant for the three wheat seedlings under the combined PEG and NaCl stress did not result in significant differences. Moreover, D reduced this value by 56%, 52%, and 26% and S decreased it by 51%, 21%, and 35% in Yannong 1212, Heng 4399, and Xinmai 19, respectively, in comparison with that in CK (Figure 5b). There were no significant differences between D and S in Yannong 1212 and Xinmai 19 (Figure 5b). Additionally, the stress types (S), different varieties (V), and S × V interaction were found to be significant for the hydraulic conductivity of the root system (Figure 5).

### 2.6. Correlation of the Root Hydraulic Conductivity with Physiological Traits in Wheat Seedlings under PEG and NaCl Stress

The results of the correlation analysis of the three wheat varieties were very similar but different (Table 4). The variables indicated that the physiological traits of the three wheat varieties were more significantly correlated with the root hydraulic conductivity per plant than with the theoretical hydraulic conductivity. In the three wheat varieties, the theoretical hydraulic conductivity had a positive association with traits such as RL, RS, RV, N−shoot, K^+^−shoot, and K^+^−root and a negative association with traits such as Na^+^−shoot and Na^+^−root. For the root hydraulic conductivity per plant, RL, N−shoot, and K^+^−shoot were significantly and positively correlated with it, and their correlation coefficient values reached 0.91–0.99 in the three wheat seedlings. The traits that had a significantly positive relationship with the root hydraulic conductivity in the three wheat varieties were RL, RS, RV, N−shoot, K^+^−shoot, N−root, and K^+^−root. However, the indexes of the extremely negative correlations of the root hydraulic conductivity in the three wheat varieties were those of Na^+^-shoot, Na^+^/K^+^−shoot, SOD−shoot, POD−shoot, Pro−shoot, SS−shoot, St−shoot, Fr−shoot, Sc−shoot, Na^+^/K^+^−root, Pro−root, SS−root, St−root, Fr−root, and Sc−root. Additionally, the numbers of indexes with correlation coefficients that were ≥0.90 or ≤−0.90 in Heng 4399 and Xinmai 19 were equivalent and greater than in Yannong 1212.

Furthermore, we explored the relation between root hydraulic conductivity and the physiological characteristics of Yannong 1212, Heng 4399, and Xinmai 19 under PEG stress, NaCl stress and coupled PEG and NaCl stress via a path analysis, for which the representative traits (root parameters: N content, NSC content, Na^+^ and K^+^ content, antioxidant enzymes, and osmotic adjustment) that had the greatest correlation coefficients with Lpr in the Pearson correlation coefficient analysis were used. As shown in Figure 6, we found that RL directly and positively affected Lpr in Yannong 1212 and Heng 4399. In Xinmai 19, RS did so. However, only the root parameters (RL or RS) positively and markedly affected Lpr in the three wheat varieties. N-shoot was positively and greatly influenced by Lpr in the wheat seedings. However, in Heng 4399, K^+^-root had a direct and extremely positive association with Lpr, while Na^+^/K^+^−root had such an association in Xinmai 19. However, SS-shoot had an significantly adverse association with Lpr in Heng 4399 and Xinmai 19. Additionally, Lpr did not actively affect K^+^-shoot in Yannong 1212.

## 3. Discussion

### 3.1. Response of Root Morphological Parameters to PEG and NaCl Stress in Wheat Seedlings

The root system is the organ of a crop that is in direct contact with soil, and when the soil moisture is insufficient and the salt content is high, the growth and development of the root system are the first to be affected; this, in turn, affects the nutrient uptake and shoot growth of the crop [42]. The results of the present study showed that drought stress and dual PEG and NaCl stress significantly reduced the root length, root surface area, and root volume of wheat seedlings, which was consistent with the results of previous studies [43]. This is mainly because the root growth rate is determined, in part, by anisotropic cell expansion, which allows root production when the division and expansion of the root system towards anisotropic cells are hindered [44,45]. However, S increased the root length, root surface area, and root volume, and it decreased the mean root diameter of Heng 4399, indicating that a moderate salt concentration promoted root growth in Heng 4399 (Figure 1a,c,d). It was also shown that salinity promoted root growth and improved the root/shoot ratio and root activity in salt-tolerant plants [46]. This response is one of the strategies used by plants to adapt to salt stress. When plants are exposed to salt stress, they enhance their salt tolerance by altering their root structure to ensure that their root system absorbs water and minerals and maintains normal growth and development [47]. Since ensuring plant survival is crucial under adverse conditions, this mechanism is also desirable. However, it was reduced in Yannong 1212 and Xinmai 19 (Figure 1a,c,d). These results also suggest that the salt tolerance of Heng 4399 is stronger than that of Yannong 1212 and Xinmai 19. Because of the changes in the root morphological parameters in response to the conditions of single or combined PEG and NaCl stress, Yannong 1212 and Xinmai 19 were less sensitive to drought than to NaCl stress (Figure 1).

### 3.2. Response of K^+^ and Na^+^ Contents to PEG and NaCl Stress in Wheat Seedlings

PEG and NaCl stress adversely affect plant growth, development, and production. PEG and NaCl stresses are mainly caused by osmotic stress and ion toxicity. In comparison with the control, in general, the K^+^ content decreased and the Na^+^ content increased in the shoots and roots of Yannong 1212, Heng 4399, and Xinmai 19 under all types of stress, and the dual PEG and NaCl stress had the greatest influence on all wheat varieties (Table 1), which was a result similar to that of Jiang et al. [48] for salt stress. This indicated the competitive interaction between K^+^ and Na^+^ and that high Na^+^ levels inhibited K^+^ uptake. Accumulation of ions (mainly Na^+^) in plant cells also leads to ion toxicity and the inhibition of various developmental processes [49,50]. Normally, one should also consider that the transpiration stream might cause ions to accumulate inside plant tissues, causing irreversible damage [51]. In comparison with the control, the K^+^ content in the shoots and roots increased under salt stress mostly slightly more than under PEG stress in all wheat varieties (Table 1). This may be because the maintenance of stable K^+^ acquisition and distribution was required for ion homeostasis during salinity stress, given that the accumulation of K^+^ in plant cells balances the toxic effects of Na^+^ accumulation [52].

### 3.3. Response of N Content and Non-Structural Carbohydrates to PEG and NaCl Stress Conditions in Wheat Seedlings

It has been widely acknowledged that carbon metabolism and nitrogen metabolism are the two most essential pathways for plant growth and productivity [53]. Starch metabolism is one of metabolic processes for carbon, where starch is broken down into fructose and sucrose. These are both non-structural carbohydrate (NSC) compounds. The metabolism of NSCs in plants can reflect the carbon supply status of plants and characterize their growth and response to external stress disturbances. In addition to morphology, many studies have highlighted the importance of nonstructural carbohydrates (NSCs) in plants during droughts [54,55,56,57]. The processes of nitrogen metabolism include nitrogen uptake, transport, amino acid metabolism, etc. Nitrogen (N) is a constituent of nucleotides and proteins, and it is required in considerable quantities to maintain a plant’s photosynthetic rate and the capacity of key enzymes [58,59,60]. PEG stress and NaCl stress destroyed these two metabolisms in various plants [61,62,63,64,65]. Drought stress decreased the starch content and salt stress increased it, but a few studies have shown that starch content increases during droughts [66], which is consistent with partial results for one or two wheat varieties (Figure 3a). Guo et al. [67] found a significant reduction in starch in perennial ryegrass after drought stress in the shoots, but not in the root tissues. In contrast to the results obtained in that study, the starch content in the shoots of the three wheat varieties significantly increased, but that in the root tissues of Heng 4399 also increased, while that in the other two varieties either decreased or did not differ under PEG stress (Figure 3a). This indicated that wheat seedlings preferentially allocated carbon for shoot growth during droughts, and they did not allocate it to the roots for storage. The results of this experiment revealed that the fructose and sucrose contents in the shoots of the three wheat varieties were significantly greater than those of the root tissues under all types of stress (Figure 3b,c). An increase in soluble sugars is often widely reported as a means of providing osmotic protection in plants under PEG stress conditions [68]. For graminoids, increasing tillering is an important way to increase shoot biomass and, especially, sugar storage, which is important for growing new tillers [69,70,71]. The recovery of growth after a drought is usually accompanied by an increase in wheat tillering [72,73]. Moreover, the specific reasons for photosynthesis or the activity and concentration of rubisco were not measured in our research and need to be further studied. As for the N content, the results of this experiment showed that the PEG stress and the dual stress of PEG and NaCl both reduced the N content in the shoots and roots of the three wheat varieties, but the salt stress had no significant effects on the content in the roots of the three wheat varieties and decreased it in the shoots of the three wheat varieties (Figure 2), which was consistent with the results of Wen et al. [74]. Water stress and salt stress both suppressed the uptake of N and interfered with the transport of N in the shoots [75,76,77,78]. Unfortunately, the NH_4_^+^ content, NO_3_^−^ content, and other related indexes of nitrogen metabolism were not determined in the current experiment. This made it difficult for us to understand the variations in nitrogen metabolism in the three varieties under different types of stress.

### 3.4. Response of Antioxidant Enzymes and Osmotic Stress to PEG and NaCl Stress Conditions in Wheat Seedlings

Excessive production of reactive oxygen species (ROS) following plant stress weakens the plant’s antioxidant system, and plants face oxidative damage under stress conditions [79,80,81,82,83]. To survive in harsh environments, plant cells initiate antioxidant defense systems. The activity of enzymatic components (SOD, POD, CAT, etc.) was appreciably stimulated under drought and salt stress, thus enhancing the antioxidant defense mechanism [83,84,85,86]. In the present research, the activity of SOD, POD, and CAT in the three wheat varieties increased or decreased to different degrees under single or dual PEG and NaCl stress conditions (Figure 4). The trends of the change in activity in Yannong 1212 and Xinmai 19 were mostly similar, but they were dissimilar to that in Heng 4399 (Figure 4). This suggested that the ability to scavenge ROS with related enzymes was similar after stress in Yannong 1212 and Xinmai 19, which are resistant to drought and salt. Additionally, under salt and PEG and NaCl stress, the enzymatic activity of Heng 4399 underwent extreme increases in the roots, but not in the shoots (Figure 4). This may be attributed to the roots of Heng 4399, which have stronger abilities to scavenge ROS with antioxidant enzymes. To mitigate increased ROS production and provide metabolic protection, plants upregulate their antioxidant systems and enhance the accumulation of osmoregulatory substances [87,88]. Adequate accumulation of osmoregulatory substances effectively maintains osmotic balance and protects critical cellular structures, such as by protecting protein and membrane integrity, under stress conditions [89]. In our experiment, in comparison with the control, the soluble protein content in the shoots and roots of the wheat changed little overall, and the soluble sugar content and proline content in the shoots and roots of the three wheat varieties showed great changes under all types of stress (Table 3). Furthermore, the dual PEG and NaCl stress remarkably increased the two contents in both the shoots and roots of Yannong 1212, Heng 4399, and Xinmai 19, and it did so the most in comparison with CK (Table 3). It was shown that proline and soluble sugars were the main osmoregulatory agents in wheat under PEG stress conditions, which was in agreement with the results of experiments conducted by Ghani et al. with cucumbers [90].

### 3.5. Effects of PEG and NaCl Stress on the Root Hydraulic Conductivity of Wheat Seedlings

Root hydraulic conductivity is not only an essential indicator of the water uptake capacity of a plant’s roots, but it also a fundamental indicator of plant function and performance. It represents the amount of water flowing through a unit of root surface area (or length) per unit pressure gradient per unit of time [91]. Several studies have shown that soil characteristics and water quality have an extreme effect on root hydraulic conductivity, and salt stress greatly reduces it [20,31,36,77,92,93,94,95,96]. In this experiment, the results showed that PEG stress, NaCl stress, and combined PEG and NaCl stress significantly reduced the theoretical hydraulic conductivity and the single-plant root hydraulic conductivity of the three wheat varieties, which was in agreement with the results of previous studies (Figure 5). Therefore, it is also a physical indicator of a water deficit [19]. Under the conditions of salt stress alone, Heng 4399 had the smallest decline in root hydraulic conductivity among the three varieties, which may have been related to its drought resistance and salt tolerance (Figure 5b), but this was probably most relevant to its salt tolerance. Furthermore, the root hydraulic conductivity positively or negatively affected most of the physiological characteristics studied in the present work (Table 4). Previous studies showed that the antioxidant enzyme activity was able to regulate the root hydraulic conductivity of maize under stress conditions [48,97,98] and that NSC was essential for hydraulic recovery [38,39]. In total, this activity was actively and strongly regulated by root traits (RL/RS) and it positively and remarkably affected the N content in Yannong 1212, Heng 4399, and Xinmai 19 under PEG and NaCl stress (Figure 6 and Figure 7).

No values were measured for the single-plant root hydraulic conductivity because the injury of the wheat seedlings due to the combined PEG and NaCl stress approximated wilting. The following hypotheses were made about the effects on root hydraulic conductivity under stress conditions: (1) stress inhibits root growth, which, in turn, affects root hydraulic conductivity [99]; (2) reduced root hydraulic conductivity affects nutrient uptake in wheat, but a sharp increase in sodium content and a sharp decrease in potassium content of wheat under salinity stress conditions break the inter-ionic balance [8,100,101]; (3) the inter-ionic balance is broken and the MDA content of membrane lipid peroxides increases, but osmoregulatory substances and the regulation of antioxidant enzymes are used to maintain the balance of MDA [102,103]; (4) stress affects the distribution of non-structural carbohydrates, the accumulation of non-structural carbohydrates affects the hydraulic recovery, and the root transport capacity affects the transfer from “source” to “reservoir” [104].

Overall, our findings in the present study are summarized in Figure 7. However, further investigations are still required to explore wheat seedlings under PEG and NaCl stress and to provide a more comprehensive analysis of root hydraulic conductivity, as well as the transpiration efficiency, stomatal conductance, water-use efficiency, aquaporins, plant–soil interactions, etc.

## 4. Materials and Methods

### 4.1. Plant Materials and Experimental Design

The experiment was conducted in a phytotron at the Qiliying Comprehensive Experimental Station of the Chinese Academy of Agricultural Sciences (35°54′ N, 113°29′ E). The area of the phytotron was 13 m^2^, the temperature was 25 °C/20 °C during the day/night, the light time was from 7:00 a.m. to 7:00 p.m., and the relative humidity was 40%–50%. The three common wheat varieties used in the trial were Heng 4399 (drought- and salt-tolerant), Yannong 1212 (moderately drought-tolerant and non-salt-tolerant), and Xinmai 19 (non-drought-tolerant and non-salt-tolerant) [105,106,107,108]. Their seeds were obtained from the Institute of Dryland Farming, Hebei Academy of Agriculture and Forestry Sciences.

Three treatments were set up in the experiment—NaCl stress (S, Hoagland solution with 0.1% NaCl, 53.33 mmol/kg), PEG stress (D, Hoagland solution with 2% PEG 6000, 38.33 mmol/kg), and coupled PEG and NaCl stress (DS, Hoagland solution with 2% PEG 6000 and 0.1% NaCl, 48.00 mmol/kg)—in addition to a control (CK, Hoagland solution only, 5.00 mmol/kg). The Hoagland solution was formulated with 5 mM KNO_3_, 5 mM Ca(NO_3_)_2_ 4H_2_O, 1 mM KH_2_PO_4_, 1 mM MgSO_4_·7H_2_O, and 1 mL·L^−1^ of micronutrients [109]. The nutrient solution was changed every 4 days. The osmotic potential of the water in the solution was measured using a 5600 dew-point osmometer (VAPRO 5600, Wescor, UT, USA).

Seeds of a uniform size and completion were selected and disinfected with 0.5% sodium hypochlorite for 25 min. Then, we washed them with deionized water and placed them evenly in Petri dishes (diameter of 9 cm) containing different types of nutrient solutions at the same volume. The Petri dishes were placed in the phytotron and incubated at 25 °C. Germination was carried out in the dark for the first 2 days, and the light was turned on after 2 days. Then, 7-day wheat seedlings with consistent growth were selected for each treatment and transferred to PVC solution culture barrels (diameter: 12 cm; height: 30 cm) in an artificially climatized room, with 4 plants per barrel (connected to an oxygen pump for ventilation). Measurements of the indicators were performed 10 days after the stress treatment (DATs).

### 4.2. Measurements and Methods

#### 4.2.1. Root Hydraulic Conductivity and Root Parameters

The root hydraulic conductivity was measured in a pressure chamber (Model 3115, Plant Moisture Equipment, Santa Barbara, CA, USA) according to the method of Gal et al. [110]. The hydraulic conductivity of a single root system (Lp) is the slope of the water flux (Jv) versus the pressure difference curve (ΔP). The Lp was calculated as follows:Lp = Jv/ΔP

The roots were scanned with an Epson V800 root scanner (Perfection V800, Shanghai, China), and the root parameters, such as the root length, root diameter, root surface area, and root volume, were measured using WinRHIZO software (Rengent Instruments Inc., Nanjing, Canada).

#### 4.2.2. Determination of the Theoretical Hydraulic Conductivity

The theoretical hydraulic conductivity was calculated according to the Hagen–Poiseuille [111] model, that is, $K_{xylem}$=$\frac{\pi }{128ƞ}\sum_{i=1}^{n}D_{i}^{4}$, where ƞ indicates the viscosity coefficient of water (0.90 × 10^−6^ KPa), D indicates the diameter of the conduit (μm), and n indicates the number of conduits. Note that the root tips with a size of 0–1 cm from four randomly selected wheat plants were measured for their theoretical hydraulic conductivity in this experiment. We used ImageJ 32 software to analyze the root diameter (µm) and duct diameter (µm).

#### 4.2.3. Measurement of the Total Nitrogen, Potassium, and Sodium in Shoots and Roots

The dry samples of the shoot and root tissues in the wheat seedlings were weighed (0.15 g) and then digested with H_2_SO_4_-H_2_O_2_. Then, this solution was used to determine the total nitrogen (N) content in the shoots and roots of the wheat plants with an AA3 flow analyzer (Seal, Norderstedt, Germany). The amounts of potassium (K^+^) and sodium (Na^+^) in the shoots and roots were measured by using a flame photometer (FP650, Shanghai, China).

#### 4.2.4. Analysis of the Content of Malondialdehyde and Osmoregulatory Substances (Soluble Sugar, Proline, and Soluble Protein)

The contents of malondialdehyde (MDA), soluble sugar, proline, and soluble protein were determined according to the corresponding kits’ instructions (MDA-2-Y, KT-2-Y, PRO-2-Y, BCAP-2-W) [112,113]. The kits were purchased from Comin Biotechnology Co., Suzhou, China. The MDA, soluble sugar, proline, and soluble protein contents were recorded at wavelengths of 532 nm, 600 nm, 620 nm, 520 nm, and 562 nm, respectively.

#### 4.2.5. Analysis of the Activity of Antioxidant Enzymes

Frozen samples with a weight of 0.5 g were used to analyze the activity of antioxidant enzymes. Firstly, these samples and a crude enzyme extract (0.1 mol/L of phosphate buffer) were ground together. Then, the homogenate was placed in a centrifuge (cence, TGL-20 M) tube and centrifuged at 10,000× *g* for 10 min at 4 °C. Finally, the supernatant was transferred into another centrifuge tube to perform an assay for the analysis of the antioxidant enzyme activity. The superoxide dismutase (SOD), peroxidase (POD), and catalase (CAT) activity was estimated according to the instructions for the kits (SOD-2-W, POD-2-Y, CAT-1-W), which were acquired from Suzhou Comin Biotechnology Co., Suzhou, China. The methods used to determine the activity of SOD, POD, and CAT were the WST-8 method, the spectrophotometric method, and ammonium molybdate colorimetry, respectively [114]. The SOD, POD, and CAT activity was measured at 450 nm, 470 nm, and 405 nm, respectively, and the details test procedures used by Ru et al. [85] were used.

#### 4.2.6. Determination of the Content of Non-Structural Carbohydrates

The contents of starch, fructose, and sucrose were determined according to the instructions for the relevant kits (DF-2-Y, GT-2-Y, ZHT-2-Y, Comin Biotechnology Co., Ltd., Suzhou, China) by using an ultraviolet–visible spectrophotometer (Shimadzu, Kyoto, Japan) [115]. Firstly, we mixed the tissue mass with the extract at a volume of 1:10 mL and then ground it into a homogenate. Then, we heated the homogenate to 80 °C for 30 min. Next, we centrifuged (cence, H1650) the homogenate at 3000× *g* for 5 min, and the supernatant was transferred into another centrifuge tube to determine the contents of fructose and sucrose according to the instructions for the kits (GT-2-Y, ZHT-2-Y). The residual precipitate was diluted to gelatinize it by using 0.5 mL of distilled water, and the corresponding reagents were added according to the instructions for the kit (DF-2-Y). Finally, the absorbance of the supernatant was measured at 620 nm to calculate the starch content.

### 4.3. Statistical Analysis

The DPS V13.5 (http://www.dpsw.cn, accessed on 16 July 2023) data processing software was used for statistical analysis, and Duncan’s method was used to test the difference at the level of *p* < 0.05. EXCEL 2016 was used for basic data processing and drawing. Analysis of variance (ANOVA) was used to compare the levels in the individual stress (S) treatments and different varieties (V). Two-way ANOVA was initially performed to test the interaction of stress and variety. The Pearson correlation coefficient (r) was used to test the correlations between indicators. Path analysis with SPSSPRO (http://www.spsspro.com, accessed on 16 July 2023) was utilized in the present work.

## 5. Conclusions

Collectively, in comparison with the control, PEG and NaCl stress severely restricted root growth, decreased the N content, appropriately increased the NSC content, disturbed the balance of carbon and nitrogen metabolism, and interfered with Na^+^ and K^+^ uptake. The antioxidant enzyme activity and osmotic adjustment were enhanced in Yannong 1212, Heng 4399, and Xinmai 19 under PEG and NaCl stress alone and in combination. Expectedly, most of the root characteristics in wheat seedlings decreased, resulting in a reduction in root water conductivity. As a result, the ability of the root system to absorb water and nutrients was reduced, and the nutrient transportation from the roots to the shoots was affected—especially the N content in the shoots—thus threatening plant growth and crop yields.

## Figures and Tables

**Figure 1 plants-12-02694-f001:**
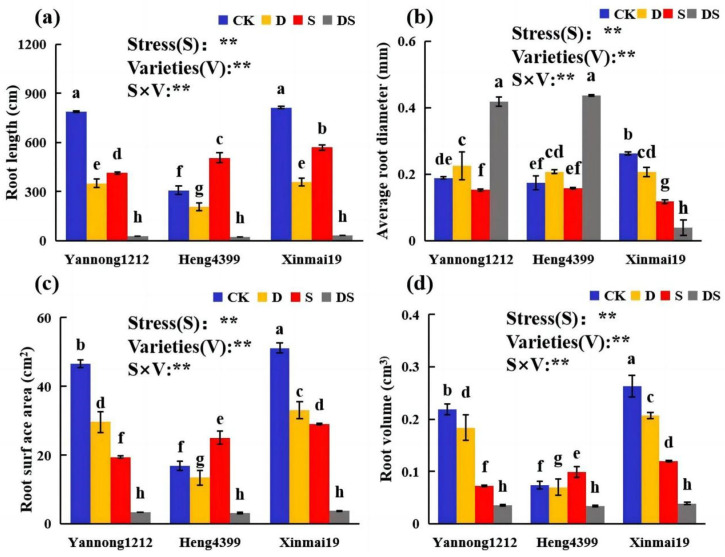
Effects of different stress treatments on wheat root length (**a**), average root diameter (**b**), root surface area (**c**), and root volume (**d**) of the different varieties. CK = control; D = PEG stress; S = NaCl stress; DS = coupled PEG and NaCl stress. The values are the means ± standard deviations (*n* = 3). Different lowercase letters indicate significant differences (*p* < 0.05). ANOVA was used to assess the stress (S), variety (V), and S × V. Stress: stress types; Variety: different varieties. ** indicates a significant difference at the level of *p* < 0.01.

**Figure 2 plants-12-02694-f002:**
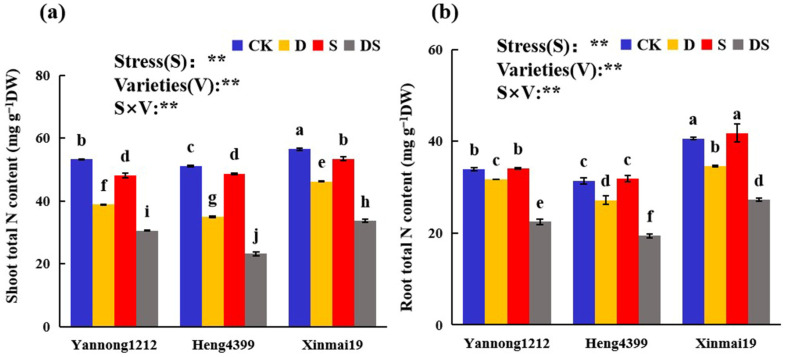
Effects of different stress treatments on the total nitrogen in the shoots (**a**) and roots (**b**) of wheat seedlings. CK = control; D = PEG stress; S = NaCl stress; DS = coupled PEG and NaCl stress. The values are the means ± standard deviations (*n* = 3). Different lowercase letters indicate significant differences (*p* < 0.05). ANOVA was used to assess the stress (S), variety (V), and S × V. Stress: Stress types, Varieties: different varieties. ** indicates a significant difference at the level of *p* < 0.01.

**Figure 3 plants-12-02694-f003:**
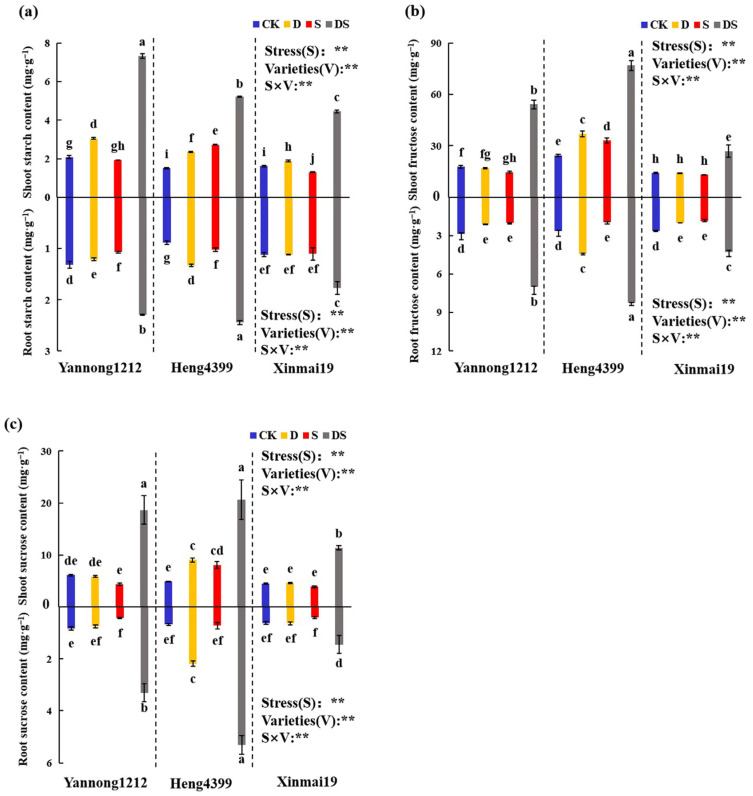
Effects of different stress treatments on non-structural carbohydrates (NSCs) in the roots and shoots of wheat seedlings. (**a**) The soluble sugar content in the shoots and roots of Yannong 1212, Heng 4399, and Xinmai 19. (**b**) The fructose content in the shoots and roots of Yannong 1212, Heng 4399, and Xinmai 19. (**c**) The sucrose content in the shoots and roots of Yannong 1212, Heng 4399, and Xinmai 19. CK = control; D = PEG stress; S = NaCl stress; DS = coupled PEG and NaCl stress. The values are the means ± standard deviations (*n* = 3). Different lowercase letters indicate significant (*p* < 0.05) differences. ANOVA was used for the stress (S), variety (V), and S × V. Stress: stress types, Varieties: different varieties. ** indicates a significant difference at the level of *p* < 0.01.

**Figure 4 plants-12-02694-f004:**
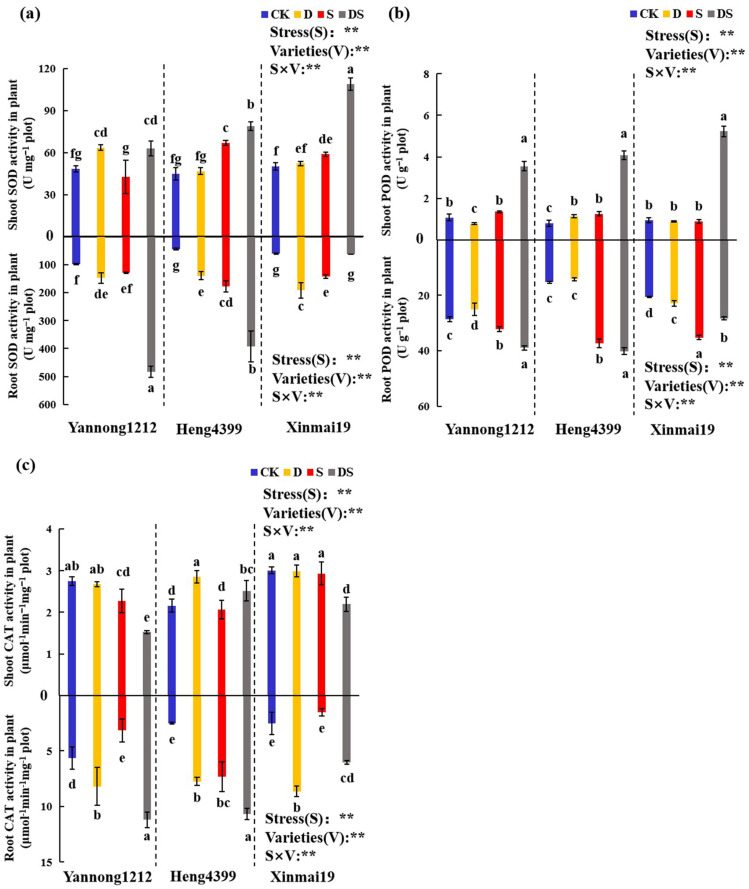
Effects of different stress treatments on antioxidative enzyme activity in the roots and shoots of wheat seedlings. (**a**) The SOD activity in the shoots and roots of Yannong 1212, Heng 4399, and Xinmai 19. (**b**) The POD activity in the shoots and roots of Yannong 1212, Heng 4399, and Xinmai 19. (**c**) The CAT activity in the shoots and roots of Yannong 1212, Heng 4399, and Xinmai 19. SOD = superoxide dismutase; POD = peroxidase; CAT = catalase; CK = control; D = PEG stress; S = NaCl stress; DS = coupled PEG and NaCl stress. The values are the means ± standard deviations (*n* = 3). Different lowercase letters indicate significant (*p* < 0.05) differences. ANOVA was used to assess the stress (S), variety (V), and S × V. Stress: stress types, Varieties: different varieties. ** indicates a significant difference at the level of *p* < 0.01.

**Figure 5 plants-12-02694-f005:**
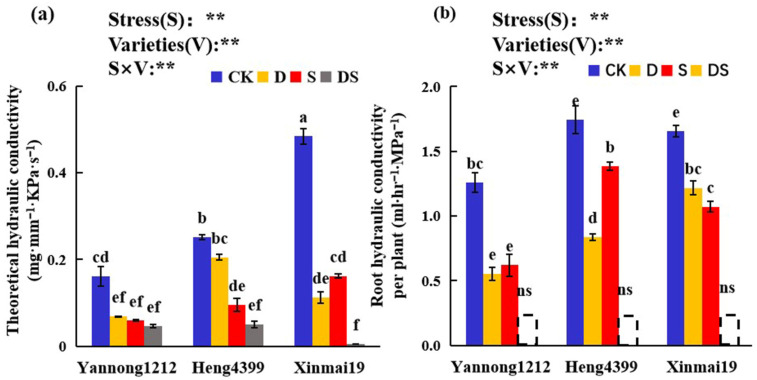
Effects of the different stress treatments on the theoretical hydraulic conductivity (**a**) and root hydraulic conductivity per plant (**b**) in the wheat seedlings. CK = control; D = PEG stress; S = NaCl stress; DS = coupled PEG and NaCl stress. The values are the means ± standard deviation (*n* = 3). Different lowercase letters indicate significant differences (*p* < 0.05); ns indicates that there was no significant difference. ANOVA was used to assess the stress (S), variety (V), and S × V. Stress: stress types, Varieties: different varieties. ** indicates a significant difference at the level of *p* < 0.01.

**Figure 6 plants-12-02694-f006:**
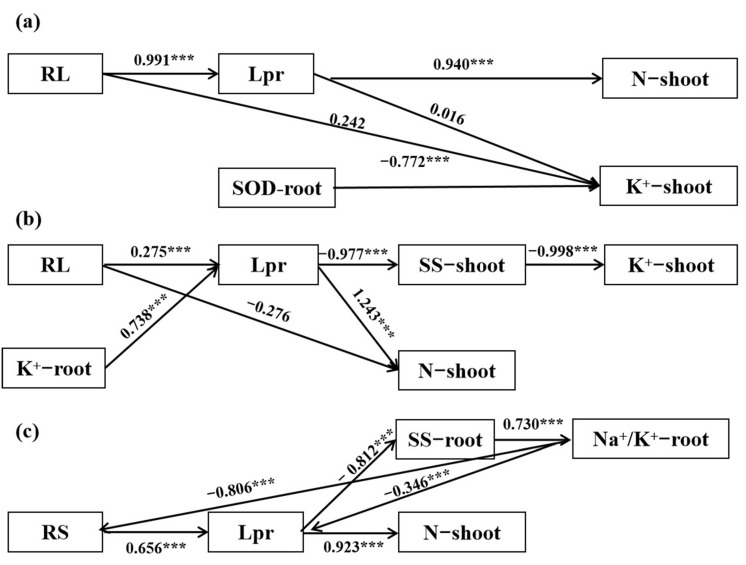
Path analysis of the relation between root hydraulic conductivity and the physiological characteristics of Yannong 1212 (**a**), Heng 4399 (**b**), and Xinmai 19 (**c**) under PEG stress, NaCl stress, and coupled PEG and NaCl stress. Lpr = root hydraulic conductivity per plant; RL = root length; RS = root surface area; N−shoot = total N content in shoots; K^+^−shoot = K^+^ content in shoots; K^+^−root = K^+^ content in roots; Na^+^/K^+^−root = Na^+^/K^+^ ratios in roots; SOD−root = SOD activity in roots; SS−shoot = soluble sugar content in shoots; SS−root = soluble sugar content in roots; *** indicates a significant difference at the level of *p* < 0.001.

**Figure 7 plants-12-02694-f007:**
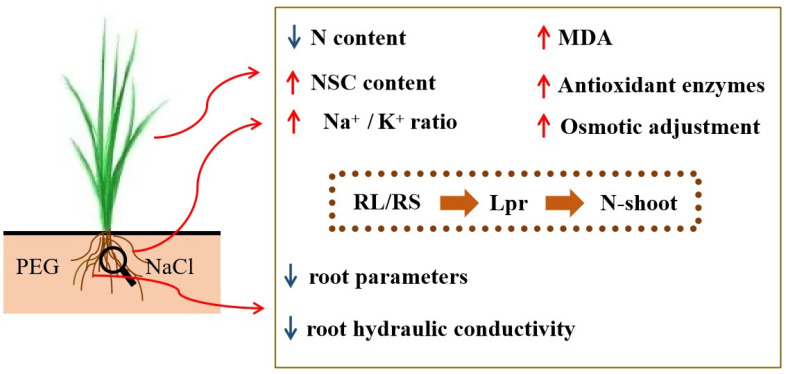
Graphical response and the relation of root hydraulic conductivity and the physiological characteristics of wheat seedlings under PEG and NaCl stress. N content = total N content in shoots and roots; NSC content = non-structural carbohydrate content in shoots and roots; Na^+^ and K^+^ content = Na^+^ and K^+^ content in shoots and roots; MDA = malondialdehyde content in shoots and roots; RL = root length; RS = root surface area; Lpr = root hydraulic conductivity per plant; N-shoot = total N content in shoots.

**Table 1 plants-12-02694-t001:** Effects of different stress treatments on the potassium and sodium contents in the roots and shoots of wheat seedlings.

Treatments		Root Ion Concentrations (mg × g^−1^ Dry Weight)				Ratios	
		K^+^		Na^+^		Na^+^/K^+^	
		Shoot	Root	Shoot	Root	Shoot	Root
Yannong 1212	CK	60.56 ± 0.10 a	57.55 ± 0.44 b	0.54 ± 0.02 g	0.36 ± 0.02 e	0.01 ± 0.001 g	0.01 ± 0.001 e
	D	49.45 ± 0.21 e	49.21 ± 0.21 c	0.56 ± 0.11 g	0.38 ± 0.00 e	0.01 ± 0.001 g	0.01 ± 0.001 e
	S	51.67 ± 0.45 d	45.95 ± 0.47 d	2.85 ± 0.02 e	3.70 ± 0.01 c	0.06 ± 0.001 e	0.08 ± 0.001 de
	DS	14.30 ± 0.05 j	8.32 ± 0.74 h	4.91 ± 0.02 b	3.63 ± 0.02 c	0.34 ± 0.002 b	0.45 ± 0.074 b
Heng 4399	CK	55.41 ± 0.36 b	50.01 ± 1.14 c	0.38 ± 0.02 gh	0.36 ± 0.02 e	0.01 ± 0.001 g	0.02 ± 0.003 e
	D	41.29 ± 0.06 g	28.54 ± 2.06 e	0.25 ± 0.02 h	0.38 ± 0.00 e	0.01 ± 0.001 g	0.04 ± 0.002 e
	S	48.17 ± 0.22 f	45.91 ± 0.53 d	4.24 ± 0.04 d	3.70 ± 0.01 c	0.06 ± 0.001 d	0.16 ± 0.003 cd
	DS	18.31 ± 0.21 i	11.33 ± 0.62 g	6.48 ± 0.05 a	3.63 ± 0.02 c	0.34 ± 0.012 a	0.58 ± 0.061 a
Xinmai 19	CK	59.84 ± 0.59 a	66.51 ± 0.23 a	0.53 ± 0.01 g	0.79 ± 0.03 de	0.01 ± 0.001 g	0.01 ± 0.001 e
	D	53.48 ± 0.10 c	51.21 ± 0.38 c	0.48 ± 0.00 g	0.83 ± 0.06 de	0.01 ± 0.001 g	0.02 ± 0.001 e
	S	54.42 ± 0.43 bc	59.20 ± 0.07 b	2.32 ± 0.03 f	3.74 ± 0.01 c	0.06 ± 0.001 d	0.06 ± 0.001 e
	DS	22.62 ± 0.10 h	19.65 ± 0.31 f	4.53 ± 0.14 c	3.58 ± 0.01 c	0.34 ± 0.012 a	0.18 ± 0.003 c
	ANOVA						
	Stress (S)	**	**	**	**	**	**
	Varieties (V)	**	**	**	**	**	**
	S × V	**	**	**	**	**	**

Note: CK = control; D = PEG stress; S = NaCl stress; DS = coupled PEG and NaCl stress. The values are the means ± standard deviations (*n* = 3). Different lowercase letters represent significant differences between the experimental treatments at *p* < 0.05. ANOVA was used for stress (S), variety (V), and S × V. Stress: stress types, Varieties: different varieties. ** indicates a significant difference at the level of *p* < 0.01.

**Table 2 plants-12-02694-t002:** Effects of PEG and NaCl stress on MDA (nmol/mg prot^−1^) in wheat seedlings.

	Yannong 1212		Heng 4399		Xinmai 19	
	Shoots	Roots	Shoots	Roots	Shoots	Roots
CK	0.39 ± 0.01 f	0.95 ± 0.02 c	0.50 ± 0.02 c	0.50 ± 0.02 g	0.46 ± 0.02 de	0.60 ± 0.02 f
D	0.39 ± 0.02 f	0.73 ± 0.01 de	0.57 ± 0.01 ab	0.76 ± 0.01 d	0.58 ± 0.01 a	0.67 ± 0.02 ef
S	0.43 ± 0.01 ef	1.18 ± 0.03 b	0.45 ± 0.02 de	0.94 ± 0.04 c	0.54 ± 0.01 b	0.90 ± 0.02 c
DS	0.47 ± 0.01 cd	0.96 ± 0.02 c	0.54 ± 0.02 b	1.37 ± 0.05 a	0.46 ± 0.01 cde	0.66 ± 0.01 ef
ANOVA						
Stress (S)	**	**	**	**	**	**
Varieties (V)	**	**	**	**	**	**
S × V	**	**	**	**	**	**

Note: CK = control; D = PEG stress; S = NaCl stress; DS = coupled PEG and NaCl stress. The values are the means ± standard deviations (*n* = 3). Different lowercase letters indicate significant (*p* < 0.05) differences. ANOVA was used for the stress (S), variety (V), and S × V. Stress: stress types, Varieties: different varieties. ** indicates a significant difference at the level of *p* < 0.01.

**Table 3 plants-12-02694-t003:** Effects of PEG and NaCl stress on the osmolytes of wheat seedlings.

		Soluble Protein		Soluble Sugar		Proline	
		(mg·g^−1^)		(mg·g^−1^)		(μg·g^−1^)	
		Shoots	Roots	Shoots	Roots	Shoots	Roots
Yannong 1212	CK	9.52 ± 0.39 b	1.65 ± 0.20 de	3.12 ± 0.07 gh	0.84 ± 0.02 f	3.06 ± 0.25 e	3.23 ± 0.15 fg
	D	9.59 ± 0.05 b	2.12 ± 0.11 cde	3.86 ± 0.11 f	0.94 ± 0.01 e	6.02 ± 0.06 e	5.10 ± 0.02 e
	S	10.30 ± 0.05 b	1.68 ± 0.16 de	2.56 ± 0.04 i	0.74 ± 0.02 g	5.02 ± 0.30 e	3.16 ± 0.25 fg
	DS	14.61 ± 1.20 a	2.17 ± 0.28 cde	11.47 ± 0.01 b	2.76 ± 0.03 b	449.31 ± 15.21 b	48.84 ± 0.30 b
Heng 4399	CK	11.18 ± 1.01 a	3.48 ± 0.63 a	3.18 ± 0.02 g	0.92 ± 0.01 e	3.31 ± 0.34 e	4.51 ± 0.49 ef
	D	8.87 ± 0.34 b	2.97 ± 0.03 ab	6.70 ± 0.08 d	2.58 ± 0.02 c	111.39 ± 1.92 d	10.46 ± 0.44 d
	S	11.00 ± 0.21 a	1.53 ± 0.01 e	5.12 ± 0.10 e	1.18 ± 0.02 d	4.25 ± 0.09 e	3.68 ± 0.37 efg
	DS	10.02 ± 0.19 ab	2.36 ± 0.12 bcd	12.54 ± 0.15 a	3.15 ± 0.05 a	178.03 ± 8.62 c	54.53 ± 1.50 a
Xinmai 19	CK	8.38 ± 0.57 ab	2.73 ± 0.14 bc	2.12 ± 0.08 j	0.66 ± 0.01 h	2.22 ± 0.11 e	2.74 ± 0.18 g
	D	7.33 ± 0.11 b	2.43 ± 0.07 bcd	2.88 ± 0.05 f	0.77 ± 0.01 g	2.67 ± 0.14 e	3.63 ± 0.02 efg
	S	7.40 ± 0.02 b	1.77 ± 0.02 de	2.08 ± 0.06 j	0.77 ± 0.05 g	2.37 ± 0.16 e	2.99 ± 0.19 fg
	DS	10.04 ± 1.04 a	3.68 ± 0.33 a	9.82 ± 0.10 c	1.15 ± 0.02 d	483.52 ± 13.89 a	25.69 ± 0.20 c
	ANOVA						
	Stress (S)	**	**	**	**	**	**
	Varieties (V)	**	**	**	**	**	**
	S × V	**	**	**	**	**	**

Note: CK = control; D = PEG stress; S = NaCl stress; DS = coupled PEG and NaCl stress. Each datum is the mean of three replicates ± standard error. Different lowercase letters in a column indicate significant (*p* < 0.05) differences. ANOVA was used for the stress (S), variety (V), and S × V. Stress: stress types, Varieties: different varieties. ** indicates a significant difference at the level of *p* < 0.01.

**Table 4 plants-12-02694-t004:** Pearson correlation analysis between the hydraulic conductivity and physiological parameters measured in the different wheat seedlings.

	Yannong 1212		Heng 4399		Xinmai 19	
Parameters	TLpr	Lpr	TLpr	Lpr	TLpr	Lpr
RL	0.85 **	0.99 **	0.82 **	0.94 **	0.87 **	0.92 **
RA	−0.42	−0.79 **	−0.56 *	−0.90 **	0.76 **	0.93 **
RS	0.83 **	0.93 **	0.95 **	0.83 **	0.84 **	0.98 **
RV	0.74 **	0.78 **	0.98 **	0.63 *	0.76 **	0.94 **
N−shoot	0.71 **	0.93 **	0.55	0.98 **	0.77 **	0.93 **
K^+^−shoot	0.62 *	0.91 **	0.73 **	0.98 **	0.68 *	0.97 **
Na^+^−shoot	−0.63 *	−0.79 **	−0.96 **	−0.64 *	−0.64 *	−0.94 **
Na^+^/K^+^−shoot	−0.51	−0.82 **	−0.83 **	−0.82 **	−0.61 *	−0.96 **
CAT−shoot	0.60 *	0.81 **	0.07	−0.54	0.54	0.87 **
SOD−shoot	−0.3	−0.56 *	−0.96 **	−0.68 *	−0.63 *	−0.96 **
POD−shoot	−0.44	−0.75 **	−0.77 **	−0.89 **	−0.55	−0.93 **
Pro−shoot	−0.45	−0.79 **	−0.47	−0.96 **	−0.57 *	−0.93 **
Sp−shoot	−0.48	−0.75 **	0.04	0.41	−0.22	−0.57 *
MDA−shoot	−0.63 *	−0.79 **	0.13	−0.51	−0.39	0.2
SS−shoot	−0.44	−0.80 **	−0.74 **	−0.98 **	−0.61 *	−0.94 **
St−shoot	−0.49	−0.83 **	−0.87 **	−0.91 **	−0.57 *	−0.90 **
Fr−shoot	−0.4	−0.76 **	−0.75 **	−0.94 **	−0.52	−0.88 **
Sc−shoot	−0.38	−0.74 **	−0.73 **	−0.92 **	−0.54	−0.91 **
N−root	0.49	0.83 **	0.56 *	0.93 **	0.66 *	0.81 **
K^+^−root	0.62 *	0.89 **	0.58 *	0.98 **	0.75 **	0.96 **
Na^+^−root	−0.65 *	−0.62 *	−0.94 **	−0.41	−0.56 *	−0.72 **
Na^+^/K^+^−root	−0.51	−0.79 **	−0.84 **	−0.83 **	−0.64 *	−0.97 **
CAT−root	−0.34	−0.69 **	−0.83 **	−0.88 **	−0.53	−0.31
SOD−root	−0.53	−0.85 **	−0.89 **	−0.90 **	−0.31	0.24
POD−root	−0.41	−0.63 *	−0.96 **	−0.5	−0.48	−0.42
Pro−root	−0.46	−0.80 **	−0.67 *	−0.91 **	−0.59 *	−0.94 **
Sp−root	−0.42	−0.54	0.67 *	0.23	−0.23	−0.60 *
MDA−root	−0.11	0.06	−0.92 **	−0.85 **	−0.27	−0.07
SS−root	−0.43	−0.79 **	−0.48	−0.96 **	−0.73 **	−0.98 **
St−root	−0.29	−0.69 **	−0.70 **	−0.96 **	−0.54	−0.89 **
Fr−root	−0.32	−0.68 *	−0.53	−0.93 **	−0.31	−0.77 **
Sc−root	−0.37	−0.74 **	−0.61 *	−0.96 **	−0.46	−0.80 **
TLpr	1	0.87 **	1	0.67 *	1	0.79 **
Lpr	0.87 **	1	0.67 *	1	0.79 **	1

Note: TLpr = theoretical hydraulic conductivity; Lpr = root hydraulic conductivity per plant; RL = root length; RA = average root diameter; RS = root surface area; RV = root volume; N−shoot = total N content in shoots; N−root = total N content in roots; K^+^−shoot = K^+^ content in shoots; K^+^−root = K^+^ content in roots; Na^+^−shoot = Na^+^ content in shoots; Na^+^−root = Na^+^ content in roots; Na^+^/K^+^−shoot = Na^+^/K^+^ ratios in shoots; Na^+^/K^+^−root = Na^+^/K^+^ ratios in roots; CAT−shoot = CAT activity in shoots; CAT−root = CAT activity in roots; SOD−shoot = SOD activity in shoots; SOD−root = SOD activity in roots; POD−shoot = POD activity in shoots; POD−root = POD activity in roots; Pro−shoot = proline content in shoots; Pro−root = proline content in roots; Sp−shoot = soluble protein content in shoots; Sp−root = soluble protein content in roots; MDA−shoot = MDA content in shoots; MDA−root = MDA content in roots; SS−shoot = soluble sugar content in shoots; SS−root = soluble sugar content in roots; St−shoot = starch content in shoots; St−root = starch content in roots; Fr−shoot = fructose content in shoots; Fr−root = fructose content in roots; Sc−shoot = sucrose content in shoots; Sc−root = sucrose content in roots; * indicates a significant difference at the level of *p* < 0.05; ** indicates significant difference at the level of *p* < 0.01.

## Data Availability

Data will be made available on request.

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
