# Peer review of "Effects of Single and Combined Drought and Salinity Stress on the Root Morphological Characteristics and Root Hydraulic Conductivity of Different Winter Wheat Varieties"

_plants, 2023, doi:10.3390/plants12142694_

Round 1
Reviewer 1 Report
I believe the data itself is correct in this manuscript. However, there is a problem with the experimental condition setting: Salt Stress (S) was 0.1% NaCl (P = -0.04 MPa). This corresponds to about 20 mMNaCl. This is a low concentration that is generally not considered salt stress. Even salt sensitive plants usually need more than 25 mMNaCl, and wheat usually needs more than 75 mM to be salt stressed.
We believe that the S condition in this experiment corresponds to a condition of slight osmotic stress, while the water-salt (DS) stress condition (P = -0.07 MPa) is simply a doubling of the intensity of water stress. Therefore, the effect of salt cannot be discussed. If authors want to discuss the synergistic effect of salt and water stress, they must compare the experimental condition of P=-0.07MPa (i.e. 4% PEG) with water stress alone and the DS condition in this paper.
English reads mostly OK.
Author Response
Point 1: I believe the data itself is correct in this manuscript. However, there is a problem with the experimental condition setting: Salt Stress (S) was 0.1% NaCl (P = -0.04 MPa). This corresponds to about 20 mM NaCl. This is a low concentration that is generally not considered salt stress. Even salt-sensitive plants usually need more than 25 mM NaCl, and wheat usually needs more than 75 mM to be salt stressed.
Response 1: Thanks for pointing out this issue.
We also acknowledge that the NaCl concentration in the experiment is low for wheat seedlings. However, in our experiment, the salt stress started from the seed germination stage, the wheat seeds have highly sensitive to salinity. The 0.1% NaCl concentration was determined based on our previous experiments. Under the salinity stress of 0.1% NaCl, the germination rate of wheat seeds was around 90%. Therefore, there is a cumulative effect and this low concentration resulted in the wheat being subjected to salt stress eventually, e.g., Yang et al. (2020) and Paul et al. (2019).
For the measurement of the root hydraulic conductivity (Lpr) of wheat seedlings, relatively healthy roots are required in this experiment. It is another reason we conducted this experiment under this slight salt stress. Accordingly, we revised this part accordingly (Lines 140 145). In addition, according to Meng and Fricke (2017), 53.33 mmol/kg (the osmotic potential of 0.1% NaCl) was converted into 0.13 MPa. Considering different treatments and different species, we provided the original osmolality measurements of the treatment solution and revised this manuscript in the “Plant materials and experimental design” (Lines 131 139).
References:
Meng D, Fricke W., 2017. Changes in root hydraulic conductivity facilitate the overall hydraulic response of rice (Oryza sativa L.) cultivars to salt and osmotic stress. Plant Physiology and Biochemistry, 113,64 77.
Paul K, Pauk J, Kondic Spika A, Ankica Kondic Spika, Grausgruber H, Allahverdiyev T,
Sass L, Vass I., 2019. Co--occurrence of Mild Salinity and Drought Synergistically Enhances Biomass and Grain Retardation in Wheat. Frontiers in Plant Science, 10. Liu Yang, Xuhua Li, Bin Hu, Min Liu, Wenbo Liu, Jinxin Li, Jing Zhang, Zifeng Wang., 2020. Physiological response of nitrogen fertilization to wheat seedling under mild salt stress. Soil and Fertilizer Sciences in China, 3, 16--22.
Point 2: We believe that the S condition in this experiment corresponds to a condition of slight osmotic stress, while the water--salt (DS) stress condition (P = --0.07 MPa) is simply a doubling of the intensity of water stress. Therefore, the effect of salt cannot be discussed. If authors want to discuss the synergistic effect of salt and water stress, they must compare the experimental condition of P=--0.07MPa (i.e. 4% PEG) with water stress alone and the DS condition in this paper.
Response 2: We thank the reviewer for pointing out this issue and giving this constructive suggestion. We have deleted the corresponding discussion section in the manuscript (Line 745).
Reviewer 2 Report
The article reports on the effects of presence of PEG and NaCl in the nutrient solution on wheat plants. I have several major remarks:
1. Authors call it water and salt stress, which is an inadequate definition. The problem is in that salt stress is inevitably accompanied by water deficit caused by osmotic component of salinity. Thus NaCl causes water stress. I advise authors to learn more about osmotic and ionic components of salt stress.
2. Furthermore, I am not happy with experimental design. When both treatments were combined in the present experiments water potential was about 2 times lower than in the case of each of them applied separately. Actually authors followed consequences of decreased water potential of the medium and it was possibly not important in which way (by adding PEG, NaCl or both of them) it was achieved.
3. I wonder, if accuracy of measurements with water potential meter was checked. Authors present rather strange data. Thus they claim that 0.1 % solution of NaCl had water potential of -0.04 MPa. This level of water potential is normally used as control and provides sufficient hydration (see, e.g. Biol Res 42: 239-248, 2009). So I found it difficult to believe that wheat growth could be inhibited under such conditions.
4. Furthermore, I calculated molarity of 0.1 % NaCl to compare it with literature data, which is mostly expressed in mM and found that it is about 20 mM. And published data tell that water potential of the medium is lowered by about 0.1 MPa by adding NaCl to yield 20 mM (see Lu and Fricke, Salt Stress—Regulation of Root Water Uptake in a Whole-Plant and Diurnal Context. Int. J. Mol. Sci. 2023, 24, 8070. https://doi.org/10.3390/ijms24098070). And -0.1 MPa is 2.5 times lower than -0.04 MPa measured in the present experiments.
5. In most researches study of the effects of PEG solution on the growth rate started from not less than 5 % (usually about 10 %) and inhibition of plant growth is achieved at concentrations of NaCl about 50-100 mM (see, e.g., Sabagh et al. (2021) Salinity Stress in Wheat (Triticum aestivum L.) in the Changing Climate: Adaptation and Management Strategies. Front. Agron. 3:661932. doi: 10.3389/fagro.2021.661932). So the choose of 2 % PEG and 20 mM (01.%) NaCl concentrations looks strange
6. Furthermore, authors did not specify, if the wheat plants used in their experiments belong to bread or durum wheat. This is most important since they differ greatly in salt resistance.
So my conclusion is that article cannot be accepted for publication. Authors should check their measurements of water potential of nutrient solutions and calculation of concentration of PEG and NaCl used in the present experiments.
Minor revision of Enhglish should be performed
Author Response
Response to Reviewer 2 Comments
The article reports on the effects of presence of PEG and NaCl in the nutrient solution on wheat plants. I have several major remarks:
Point 1: Authors call it water and salt stress, which is an inadequate definition. The problem is in that salt stress is inevitably accompanied by water deficit caused by osmotic component of salinity. Thus NaCl causes water stress. I advise authors to learn more about osmotic and ionic components of salt stress.
Response 1: Thanks for the valuable comments. We have gladly accepted these comments and have revised the phrase ‘water and salt stress’ with ‘drought and
salinity stress’ in the manuscript (Lines 17-19, 22 and 34; Lines 102, 110-111, and 117; Lines 132-133; Lines 220, and 222; Lines 247, and 251; Lines 267, and 273; Lines 301-303; Lines #346, and 353; Lines 385, and 387; Lines 478-479; Lines 494-495; Lines 575, 577, 583, and 589; Lines 610-612, 615, 623, and 628; Lines 750, 754, and 758).
Point 2: Furthermore, I am not happy with experimental design. When both treatments were combined in the present experiments water potential was about 2 times lower than in the case of each of them applied separately. Actually authors followed consequences of decreased water potential of the medium and it was possibly not important in which way (by adding PEG, NaCl or both of them) it was achieved. Response 2: Thanks. According to Meng and Fricke (2017), we converted the unit of osmotic pressure, mmol/kg, into MPa before. Considering different treatments and different species, we provided the original osmolality measurements of the treatment solution and revised this manuscript in the “Plant materials and experimental design” (Lines 131-139).
References:
Meng D, Fricke W., 2017. Changes in root hydraulic conductivity facilitate the overall hydraulic response of rice (Oryza sativa L.) cultivars to salt and osmotic stress. Plant Physiology and Biochemistry, 113,64-77.
Point 3: I wonder, if accuracy of measurements with water potential meter was checked. Authors present rather strange data. Thus they claim that 0.1 % solution of NaCl had water potential of --0.04 MPa. This level of water potential is normally used as control and provides sufficient hydration (see, e.g. Biol Res 42: 239--248, 2009). So I found it difficult to believe that wheat growth could be inhibited under such conditions.
Response 3: Thanks. Firstly, we have revised the water osmotic potential of the solution in the part accordingly (Lines 131-139). In this experiment, we measured the osmotic potential (mmol/kg) of NaCl and PEG solution, then converted it into MPa. According to Meng and Fricke (2017), the water potential of 2% PEG and 0.1% NaCl is -0.09 and -0.13 MPa, respectively. And we amended the part accordingly (Lines 140-145).
References:
Meng D, Fricke W., 2017. Changes in root hydraulic conductivity facilitate the overall hydraulic response of rice (Oryza sativa L.) cultivars to salt and osmotic stress. Plant Physiology and Biochemistry, 113,64-77.
Point 4: Furthermore, I calculated molarity of 0.1 % NaCl to compare it with literature data, which is mostly expressed in mM and found that it is about 20 mM. And published data tell that water potential of the medium is lowered by about 0.1 MPa by adding NaCl to yield 20 mM (see Lu and Fricke, Salt Stress——Regulation of Root Water Uptake in a Whole--Plant and Diurnal Context. Int. J. Mol. Sci. 2023, 24, 8070. https://doi.org/10.3390/ijms24098070). And --0.1 MPa is 2.5 times lower than --0.04 MPa measured in the present experiments.
Response 4: Thanks. we have revised the water osmotic potential of the solution in the part accordingly (Lines 131--139).
Point 5: In most researches study of the effects of PEG solution on the growth rate started from not less than 5 % (usually about 10 %) and inhibition of plant growth is achieved at concentrations of NaCl about 50--100 mM (see, e.g., Sabagh et al. (2021) Salinity Stress in Wheat (Triticum aestivum L.) in the Changing Climate: Adaptation and Management Strategies. Front. Agron. 3:661932. doi: 10.3389/fagro.2021.661932). So the choose of 2 % PEG and 20 mM (01.%) NaCl concentrations looks strange.
Response 5: Thanks. In the experiments, 2 % PEG, 0.1 % NaCl, and water-salt coupled stress were determined based on our previous experiment. According to Meng and Fricke (2017), the water potential of 2% PEG, 0.1% NaCl, and the coupled stress is -0.09, -0.13, and -0.12 MPa, respectively.
Point 6: Furthermore, authors did not specify, if the wheat plants used in their experiments belong to bread or durum wheat. This is most important since they differ greatly in salt resistance.
Response 6: Thanks. We agree that we did not specify if Yannong 1212, Heng 4399, and Xinmai 19 belong to bread or durum wheat. But we cited the relevant literature on the characteristics of these three wheat varieties, e.g. Sheng et al., 2013; Cao et al., 2017; Wang et al., 2022b; Xiao et al., 2018 (Line #128-129). And from these references, we have a knowledge of the three wheat varieties which are mainly widely cultivated in Hebei and Henan and belong to the common wheat. Apparently, the wheat plants used in the experiments belong to bread wheat. Accordingly, we have made corresponding revisions in the manuscript (Line #126).
Reviewer 3 Report
The article presents a very interesting work that clearly demonstrates the importance and relevance of the topic under study. The authors described in detail the research methods, data analysis and results, which makes the article understandable and reliable.
In my opinion, the article has a high scientific value.
In addition, the research may contribute to the development of new methods of tillage and breeding of plant varieties that can adapt to difficult growing conditions, tolerant of drought and salinity. All this makes the article an important contribution to the scientific field and can lead to useful practical results.
However, there are some minor errors, for example, in line 89 the word "Several" begins with a capital letter, there are errors in the bibliography, numbers 2 and 53.
There is also a wish for figures 3 and 4: to make the font clearer and larger, and it is also possible to divide it into several graphs.
Author Response
Response to Reviewer 3 Comments
The article presents a very interesting work that clearly demonstrates the importance and relevance of the topic under study. The authors described in detail the research methods, data analysis and results, which makes the article understandable and reliable.
In my opinion, the article has a high scientific value.
In addition, the research may contribute to the development of new methods of tillage
and breeding of plant varieties that can adapt to difficult growing conditions, tolerant of drought and salinity. All this makes the article an important contribution to the scientific field and can lead to useful practical results.
Point 1: However, there are some minor errors, for example, in line 89 the word "Several" begins with a capital letter, there are errors in the bibliography, numbers 2 and 53.
Response 1: Thanks. We checked this point carefully throughout the manuscript. Accordingly, we have made revisions in the corresponding places of the manuscript.
Point 2: There is also a wish for figures 3 and 4: to make the font clearer and larger, and it is also possible to divide it into several graphs.
Response 2: Thanks. We revised Figures 3 and 4 according to your suggestion.
Round 2
Reviewer 1 Report
Revision and author's explanation are acceptable.
Author Response
Thank you for your approvement.
In order to improving the quality of our manuscript, we had recheck all references to confirming that these are relevant to the contents of the manuscript. And we also had revised the format of references in the whole manuscript. Furthermore, we had utilized the editorial services of proper English language, grammar, punctuation, spelling, and overall style to improve the manuscript, which was provided by MDPI.

Reviewer 2 Report
Authors tried to improve the article according to my recommendations. But in certain respects revised article is not better than original. The term waterdrought is awful. Did authors meet such a definition anywhere? By the way, drought means absence of rains. I advise to go through the text and substitute this "waterdrought" with something like "drought imitated by additions of PEG to the nutrient solution" (at the beginning). And below terms PEG- and NaCl-treatment would clarify the design. If authors manage with this, article may be accepted. Otherwise, I recommend to reject it.
I am not qualified to judge about the quality of English
Author Response
Response: We highly appreciate the illuminating and valuable comment and suggestions given by the reviewer, which has helped us to greatly improve the quality of our manuscript.
We carefully analyzed the reviewer’s comment and had gladly accepted the suggestion. Besides, examining the entire manuscript carefully and critically, we have revised the phrase ‘drought and salinity stress’ with ‘PEG treatment and NaCl treatment’ in the corresponding manuscript. (Lines 13-18 and 30; Lines 103 and 115-117; Lines 203-207; Lines 232- 241; Lines 256; Lines 280-291; Lines 335; Lines 344, and 352; Lines 426, and 430; Lines 436; Lines 476-494; Lines 504-505, 511, and 530-531; Lines 563-564, 571, 577, 582, and 599-600; Lines 601-652, 661-699, and 712-737)

Round 3
Reviewer 2 Report
Authors addressed all my comments and I am satisfied with the results
English is not bad.
Author Response
We highly appreciate your recognition of our revisions. Besides, we had carefully carried out further calibration and revision to improve the manuscript quality.